# The Influence of Adipocyte Secretome on Selected Metabolic Fingerprints of Breast Cancer Cell Lines Representing the Four Major Breast Cancer Subtypes

**DOI:** 10.3390/cells12172123

**Published:** 2023-08-22

**Authors:** Carla Luís, Bárbara Guerra-Carvalho, Patrícia C. Braga, Carla Guedes, Emília Patrício, Marco G. Alves, Ruben Fernandes, Raquel Soares

**Affiliations:** 1Biochemistry Unit, Biomedicine Department, FMUP—Faculty of Medicine, University of Porto, 4200-450 Porto, Portugal; carlaluis@med.up.pt; 2i3S—Instituto de Investigação e Inovação em Saúde, Universidade do Porto, 4200-135 Porto, Portugal; 3Clinical and Experimental Endocrinology, UMIB—Unit for Multidisciplinary Research in Biomedicine, ICBAS—School of Medicine and Biomedical Sciences, University of Porto, 4050-313 Porto, Portugal; barbaraggcarvalho@gmail.com (B.G.-C.); patriciacbraga.1096@gmail.com (P.C.B.); alvesmarc@gmail.com (M.G.A.); 4ITR—Laboratory for Integrative and Translational Research in Population Health, University of Porto, 4099-002 Porto, Portugal; 5Laboratory of Physiology, Department of Imuno-Physiology & Pharmacology, ICBAS—School of Medicine and Biomedical Sciences, University of Porto, 4050-313 Porto, Portugal; 6LAQV-REQUIMTE, Department of Chemistry, University of Aveiro, 3810-193 Aveiro, Portugal; 7Department of Pathology, Faculty of Medicine, University of Porto, 4200-319 Porto, Portugal; 8T-BIO—Center for Translational Health and Medical Biotechnology Research, 4200-072 Porto, Portugal; carlaguedes015@gmail.com; 9Department of Clinical Pathology, São João Hospital Centre, 4200-319 Porto, Portugal; emilia.patricio@chsj.min-saude.pt; 10FCS/HEFP/UFP—Faculty of Health Sciences, Fernando Pessoa University, Fernando Pessoa Teaching Hospital, 4249-004 Porto, Portugal; ruben.fernandes@ufp.edu.pt

**Keywords:** breast cancer, obesity, molecular subtype, cancer cell metabolism, Warburg effect, cell culture

## Abstract

Molecular subtype (MS) is one of the most used classifications of breast cancer (BC). Four MSs are widely accepted according to receptor expression of estrogen, progesterone, and HER2. The impact of adipose tissue on BC MS metabolic impairment is still unclear. The present work aims to elucidate the metabolic alterations in breast cancer cell lines representing different MSs subjected to adipocyte associated factors. Preadipocytes isolated from human subcutaneous adipose tissue were differentiated into mature adipocytes. MS representative cell lines were exposed to mature adipocyte secretome. Extracellular medium was collected for metabolomics and RNA was extracted to evaluate enzymatic expression by RT-PCR. Adipocyte secretome exposure resulted in a decrease in the Warburg effect rate and an increase in cholesterol release. HER2+ cell lines (BT-474 and SK-BR-3) exhibited a similar metabolic pattern, in contrast to luminal A (MCF-7) and triple negative (TN) (MDA-MB-231), both presenting identical metabolisms. Anaplerosis was found in luminal A and TN representative cells, whereas cataplerotic reactions were likely to occur in HER2+ cell lines. Our results indicate that adipocyte secretome affects the central metabolism distinctly in each BC MS representative cell line.

## 1. Introduction

Breast cancer (BC) is a complex and heterogeneous pathology with several classifications such as pathological and clinical tumor stage, histological type, differentiation grade, and molecular subtype (MS). Each of these classifications impacts prognosis and treatment strategies. One of the most influent classifications is the MS which includes the expression assessment of estrogen receptor (ER), progesterone receptor (PR), and human epidermal growth factor receptor 2 (HER2). The stratification of receptor status distinguishes four main MSs of BC: luminal A, luminal B, HER2 positive (HER2+), and triple negative (TN) [1]. This is a decisive classification since treatment options highly depend on targeting the receptor. Luminal A and B share the common feature of being positive for both estrogen and progesterone receptor. Luminal A is characterized by high expression of ER and low expression of proliferation genes, conferring it a good prognosis and several therapeutical alternatives. On the other hand, luminal B usually expresses decreased levels of ER as well as PR when compared to luminal A, elevated levels of proliferation markers (such as Ki67) and of cell cycle-associated genes (such as G2/mitotic-specific cyclin B1—CCNB1, and MYB Proto-Oncogene Like 2—MYBL2) and may or not express HER2 [2]. HER2+ subtype is defined by the overexpression of HER2 (Neu/ErbB-2), which increases proliferation, modulates differentiation, and decreases apoptosis. It was previously associated with a poor prognosis, but new therapeutical agents have been introduced, improving the clinical outcomes of HER2+ patients [3]. TN subtype, also known as basal-like, does not express any of these receptors, which consequently leads to an unfavorable prognosis due to lack of targeted therapy. TN MS is usually associated with genetic instability for frequent mutations on TP53 gene and inactivation of the Rb pathways [4].

Adipose tissue accumulation is associated with ER and PR positive BCs and with worse prognosis [5]. Such accumulation leads to a metabolic impairment, responsible for a low chronic inflammation state, altered adipokine profile, extracellular matrix remodeling, and hormone oscillations [6], namely estrogen. Estrogen biosynthesis by aromatase, an enzyme expressed by the adipose tissue, is proposed to be the potential link, responsible for the increased BC risk in postmenopausal women [7]. Estrogens are responsible for several genomic and non-genomic mitogenic and mutagenic mechanisms, namely, DNA damage, oxidative stress, and high proliferative rates [5].

To accomplish these proliferative needs, cells shift their metabolism towards carbohydrate anabolic pathways and to the well-established Warburg effect, for nucleotides and NADPH biosynthesis, amino acids requirements, and basic energetics [8]. In the last decade, several deviations to the Warburg effect have been observed, namely the Warburg effect inversion [9]. This observation revealed, using cell-based and metabolomic assays, that under adipocyte secretome exposure, luminal A representative breast tumor cells (MCF-7) displayed an increased consumption of lactate and accumulation of glucose in the extracellular medium. Such an event was concomitant with an increase in cell proliferation, viability, and motility. These preliminary results paved the way to a more extensive investigation regarding the effect of adiposity conditions in BC metabolism, with all MSs included. The present study aims to analyze the effect of adipocyte secretome in BC by addressing enzymatic and metabolomic profiles of four different breast cancer cell lines representing the different MSs.

## 2. Materials and Methods

### 2.1. Ethical Approval

Adipose tissue was obtained in São João Hospital Centre, Porto, Portugal (CHUSJ) according to the protocol submitted to the Ethics committee of CHUSJ with the reference n. 226/2020. All subjects gave their informed consent. Procedures were executed in accordance with the international recommendations of Good Clinical Practice, the Declaration of Helsinki, and the applicable national legislation.

### 2.2. Adipocyte Differentiation

Subcutaneous adipose tissue was mechanically degraded and digested with collagenase (Sigma-Aldrich, St. Louis, MO, USA) at 37 °C for 1.5 h. The stromal vascular fraction was washed and filtered to obtain isolated preadipocytes, which were cultured at a 5% CO_2_ atmosphere at 37 °C on Dulbecco’s Modified Eagle Medium/Nutrient Mixture F12 (DMEM/F12, Sigma-Aldrich, St. Louis, MO, USA) supplemented with 17.5 mM of glucose (Sigma-Aldrich, St. Louis, MO, USA), 1% penicillin/streptomycin (Invitrogen Life technologies, Carlsbad, CA, USA), 18 µM of pantothenic acid (Sigma-Aldrich, St. Louis, MO, USA), 100 µM of ascorbic acid (Sigma-Aldrich, St. Louis, MO, USA), 16 µM of biotin (Sigma-Aldrich, St. Louis, MO, USA), and 10% newborn calf serum (NCS, Gibco, Grand Island, NY, USA). When preadipocytes reached confluency, they were differentiated into adipocytes. Cells were incubated for 3 days with DMEM/F12 medium supplemented with 17.5 mM of glucose, 1% penicillin/streptomycin, 18 µM of pantothenic acid, 100 µM of ascorbic acid, 16 µM of biotin, 3% NCS, 10 µg/mL of insulin (Sanofi, Gentilly, France), 0.1 μM of dexamethasone (Sigma-Aldrich, St. Louis, MO, USA), 0.5 mM of 3-isobutyl-methyl-xanthine (Sigma-Aldrich, St. Louis, MO, USA), and 1 μM of rosiglitazone (Sigma-Aldrich, St. Louis, MO, USA). Cells were incubated with DMEM/F12 medium supplemented with 17.5 mM of glucose, 1% penicillin/streptomycin, 18 µM of pantothenic acid, 100 µM of ascorbic acid, 16 µM of biotin, 3% NCS, 10 µg/mL of insulin, and 0.1 μM of dexamethasone until the fully differentiated phenotype was reached, at day 9. Cell medium was changed every 2/3 days. The pool of adipocyte secretomes was collected from 7 assays performed from adipose tissue samples of different patients. The purpose of this approach was to control individual variability. Secretome was preserved in −80 °C until used as treatment. Photographs were taken for each time-point using Nikon TMS with attached camera Nikon D40× (Nikon, Tokyo, Japan). 

### 2.3. Adiponectin Quantification

Adiponectin was used as an obesity-related biomarker in the differentiation of adipocytes [10]. Adiponectin was measured through a commercial ELISA kit to access differentiation grade (Cat. No. #KE00080, Proteintech, Rosemont, IL, USA). The assay was performed according to the manufacturer instructions.

### 2.4. Cell Culture

A representative breast cancer cell line of each molecular subtype was used according to the specifications described in Table 1. MCF-7, SK-BR-3, and MDA-MB-231 were obtained from the department cell biobank previously obtained from a commercial culture collection. BT-474 was obtained from i3S cell biobank.

MCF-7 (HTB-22™, ATCC, Manassas, VA, USA), BT-474 (authenticated in i3S), SK-BR-3 (HTB-30^TM^, ATCC, Manassas, VA, USA), and MDA-MB-231 (CRM-HTB-26™, ATCC, Manassas, VA, USA) were cultured in DMEM (Sigma-Aldrich, St. Louis, MO, USA), supplemented with 10% Fetal Bovine Serum (FBS, Invitrogen Life Technologies, Carlsbad, CA, USA) or 20% in BT-474 cell line, 1% penicillin/streptomycin (Invitrogen Life Technologies, Carlsbad, CA, USA), and 3.7 g/L sodium bicarbonate (Sigma-Aldrich, St. Louis, MO, USA) and maintained at 37 °C in a humidified atmosphere containing 5% CO_2_. Serum-free DMEM was used as control and treatments were performed using the secretome of differentiated adipocytes for 24 h. The nutrient composition of both cell media is presented in Appendix A. Cell assays were performed in quintuplicate.

### 2.5. NMR (Nuclear Magnetic Resonance)-Based Metabolomics

After treatment with the secretome of differentiated adipocytes, culture media was collected and analyzed by Proton Nuclear Magnetic Resonance (^1^H-NMR) for metabolite quantification. Samples were prepared by adding 45 µL of sodium fumarate (final concentration of 2 mM) in deuterated water solution to 180 µL of culture media. The samples were loaded into 3 mm NMR tubes (Norell, Landisville, NJ, USA) and analyzed using the NOR5X3INSB optimizer inserts (Norell, Landisville, NJ, USA). Solvent-suppressed 1D ^1^H-NMR spectra were acquired on a Bruker Avance III HD 500 MHz spectrometer equipped with a 5-mm TXI probe (Bruker Corporation, Billerica, MA, USA), at 298 K. A zgpr pulse sequence was used and a total of 16 scans were acquired for each sample. After the acquisition, ^1^H-NMR spectra were processed, and chosen metabolite peaks were integrated using the NUTS-Pro NMR software (Acorn NMR, Inc., Fremont, CA, USA). Sodium fumarate was used as an internal reference (6.50 ppm) for the quantification of the following metabolites (multiplicity, chemical shift): valine (doublet, 0.97 ppm), isoleucine (doublet, 1.00 ppm), alanine (doublet, 1.46 ppm), acetate (singlet, 1.90 ppm), glutamate (multiplet, 2.35 ppm), pyruvate (singlet, 2.35 ppm), glutamine (multiplet, 2.44 ppm), lactate (quartet, 4.10 ppm), and glucose (doublet, 5.22 ppm). Results were normalized to β-actin total RNA (Appendix A). Metabolite consumption or release was determined for each condition by the difference between pre and post treatment.

### 2.6. Triglycerides and Cholesterol Quantification

Triglyceride quantification was determined by enzymatic colorimetric assay (Triglycerides reagent, Ref #OSR66118, Beckman Coulter^®^, Brea, CA, USA), on the AU 5800 analyzer (Beckman Coulter^®^, Brea, CA, USA) according to the manufacturer instructions and São João Hospital Center patronized protocol. Total cholesterol concentration was determined by an enzymatic, colorimetric method through a commercial kit (Liquick Cor-CHOL, Ref #2-205, Cormay, Varsóvia, Poland) according to the manufacturer protocol. Results were normalized to β-actin total RNA and presented as the difference between pre and post treatment.

### 2.7. RNA Extraction

After treatment, cells were suspended in RNA later (Invitrogen Life technologies, Carlsbad, CA, USA), and maintained at −20 °C. Afterwards, RNA later was removed by centrifugation. Cells were then washed and resuspended in 500 µL PBS with 7 µL of Poly(A) (Merck, Darmstadt, Germany). RNA extraction was performed using the Lab-Aid 824 s Nucleic Acid Extraction System (Zeesan, Xiang’an, China). Total RNA was quantified using Multiskan SkyHigh Microplate Spectrophotometer (Thermo Fisher Scientific, Waltham, MA, USA). Total RNA concentration was standardized to 5 µg/mL.

### 2.8. RT-PCR

Real time PCR was performed in sealed 96-well microplates using qTOWER^3^ instrument (Analytik Jena, Jena, Germany) and a SYBR Green commercial kit (SensiFAST SYBR No-ROX One-Step MIX, Meridian Bioscience Inc., Cincinnati, OH, USA). Primer sequences (Sigma-Aldrich, St. Louis, MO, USA) are described in Table 2, β-actin was used as housekeeping gene and enzyme isoform choice according to the literature [12]. PCR conditions were as follows: initial denaturation for 2 min at 95 °C, 40 cycles of 5 s at 95 °C, and 30 s at 60 °C. RT-PCR assays were performed in duplicate of each 5 cell replicates. Results were presented as fold increase of the control.

### 2.9. Statistical Analysis

Results were expressed as mean ± SD. Data were analyzed with GraphPad Prism 6.0 (GraphPad Software Inc., San Diego, CA, USA). Adipocyte differentiation was evaluated by ANOVA test. Metabolomic, lipid, and PCR results were evaluated by Student’s *t*-test between controls and treatments. Significance was set at *p* < 0.05.

## 3. Results

### 3.1. Adipocyte Differentiation

Figure 1a–d displays the phenotypic alterations of adipocyte differentiation from day 3 to day 9. The adipocyte differentiation rate was quantified using adiponectin ELISA kit. Day 9 exhibited the highest differentiation rate (Figure 1e). We then used day-9 secretome in the next experiments. From day 9 onwards, no significant phenotypic differences were observed in the differentiation rate.

### 3.2. Carbohydrate Metabolic Profile

The levels of glucose as well as glycolysis end products, pyruvate, and their dual metabolites, lactate, and acetate are presented in Figure 2.

Under adipocyte secretome exposure, MCF-7 did not achieve significant alterations in glucose consumption nor in lactate and acetate production. Yet, our results show a decreasing trend for both lactate and acetate production when subjected to adipocyte secretome. Pyruvate variation was also not significantly affected by adiposity; however, our results point to a shift in pyruvate utilization from a consumption in the control condition, to a release under the influence of adipocyte secretome, although with a high standard deviation.

When compared to the other MS cell lines, luminal B representative cell line (BT-474) revealed lower levels of these four metabolites. Subjected to adipocyte secretome, these cells exhibited a significant increase in acetate consumption (*p*-value = 0.024).

Regarding SK-BR-3 cell line representing luminal B MS, no significant differences were observed in any of the metabolites, although there was a tendential decrease in glucose consumption followed by a decrease in pyruvate and lactate production under adiposity condition. The switch from acetate production, observed in the control condition, towards acetate consumption under adiposity environment was a noteworthy feature.

Finally, MDA-MB-231 cell line representing triple negative MS showed a significant decrease in both glucose consumption and lactate production under adiposity when compared to control. Under exposure to adipocyte secretome, these cells did not depend much on glucose uptake (*p*-value = 0.0213), resulting in a decrease in lactate production (*p*-value < 0.0001), implying that the Warburg effect in this aggressive cell line slowed down. A significant switch from acetate production in control conditions towards acetate consumption in adiposity conditions was also found (*p*-value = 0.0133). Altogether, these findings suggest that under exposure to secretome from adipocyte differentiation, the cell lines present distinct metabolic pathways.

### 3.3. Amino Acid Metabolic Pattern

Amino acids are used as protein building blocks as well as energy fuel in stress conditions. Figure 3 presents the profile of alanine, glutamine, glutamate, isoleucine, and valine levels in culture media.

Alanine and pyruvate, for instance, can be directly interconverted. When exposed to adipocyte secretome, alanine metabolism was significantly altered in MCF-7, SK-BR-3, and MDA-MB-231 cell lines. Alanine shifted from production to consumption in MCF-7 (*p*-value = 0.0010) and MDA-MB-231 cell lines (*p*-value < 0.0001), while in SK-BR-3 cells (*p*-value = 0.0366) and BT-474, alanine release was reduced.

Glutamine consumption was significantly decreased in luminal A, luminal B, and TN representative cell lines when exposed to adipocyte secretome (*p*-value MCF-7 = 0.0003; *p*-value SK-BR-3 = 0.0097; *p*-value MDA-MD-231 = 0.0003). No significant difference was found in HER2+ representative cell line. This amino acid is directly converted into glutamate by glutaminase. Therefore, we next examined glutamate levels in our experiment.

Glutamate production in adipocyte secretome condition decreases significantly in luminal A (*p*-value = 0.0377) and TN (*p*-value = 0.0495), accompanying the decrease in glutamine consumption levels observed. Interestingly, glutamate release increases significantly in luminal B (*p*-value = 0.0167) and HER2+ (*p*-value = 0.0464) representative cell lines.

The role of branched amino acids, such as valine and isoleucine, is not clear in BC progression. We next investigated the level of these two amino acids in our study. No significant difference was found in any of the cell lines except for MDA-MB-231 TN cells, where these two amino acids were significantly less consumed after exposure to adipocyte secretome (*p*-value isoleucine < 0.0001; *p*-value valine = 0.0052).

### 3.4. Lipid Characterization

Triglycerides (TG) and cholesterol levels in culture media were then addressed (Figure 4).

No differences were found in TG levels under adipocyte secretome exposure when compared to control, although a slight increase was observed in MCF-7 and MDA-MB-231.

In contrast, a significant increase in cholesterol release was observed in every MS representative cell line when subjected to adipocyte secretome (*p*-value MCF-7 = 0.0179; *p*-value BT-474 = 0.0301; *p*-value SK-BR-3 = 0.0086; *p*-value MDA-MD-231 = 0.0005).

### 3.5. Glycolytic and Gluconeogenic Enzymatic Fingerprints

To complement metabolomic data, we further examined glycolytic and gluconeogenic enzyme expression in the different cell lines subjected to adipocyte secretome (Figure 5).

A distinct fingerprint for each MS can be visualized. BT-474 did not reveal any significant alteration in the expression pattern of both glycolysis and gluconeogenesis enzymes between control and treatment. In contrast, MCF-7 presented an increased expression of Phosphofructokinase (PFK) and a decreased expression of Hexokinase (HK), and of the enzymes that catalyze the late steps of gluconeogenesis: Fructose-1,6-bisphosphatase (FBP), and Glucose-6-phosphatase (G6P).

In contrast to MCF-7 fingerprint, SK-BR-3 displayed significantly increased levels of glycolytic and gluconeogenic enzymes: HK, Pyruvate kinase (PK), Phosphoenolpyruvate carboxykinase (PCK), FBP, and G6P. We also observed a tendency towards increased Pyruvate Carboxylase (PC) expression, but statistical significance was not achieved. PFK expression was not altered in SK-BR-3 cells.

Triple negative MDA-MB-231 representative cell line enzyme expression profile was similar to MCF-7, with a significant decrease in HK, FBP, and G6P and a decreasing trend in all other enzymes except PFK, which is increased under adipocyte secretome exposure.

### 3.6. TIGAR Expression Pattern

Given the results obtained in PFK expression, we further quantified the expression of TP53 Induced Glycolysis Regulatory Phosphatase (TIGAR), an enzyme that modulates glycolysis by targeting PFK. Results are illustrated in Figure 6. TIGAR was significantly downregulated in MCF-7 in adiposity conditions. A similar pattern was observed in MDA-MB-231 cells, though without reaching statistical significance. Inversely, TIGAR expression was upregulated in BT-474 and in SK-BR-3, being statistically significant in SK-BR-3.

## 4. Discussion

Carbohydrate metabolism and the Warburg effect in BC have been widely and extensively studied. However, the metabolic associated pathways are far from being fully understood. Tumor cell plasticity and the ability to use multiple metabolic pathways to increase survival advantage is acknowledged and recognized [13]. Recently, several deviations to the Warburg effect have been described such as the Reverse Warburg effect [14] or the Warburg effect inversion [9]. BC MSs present different metabolic signatures namely associated to the tricarboxylic acid (TCA) cycle, amino acid metabolism, and glycolysis/gluconeogenesis [15]. The present study aimed to characterize the metabolic profile alterations under adiposity conditions of cell lines representing the four MS.

Overall, we found three main similarities across the different molecular subtypes when submitted to adipocyte secretome. Firstly, there was a decrease in lactate production, which led us to postulate that the Warburg effect is attenuated by adipocyte released factors. One possible trigger for this scenario is the presence of high levels of fatty acids in the adipocyte secretome, which feed the TCA cycle, providing a robust energy source to tumor cells. A mechanism similar to the “glucose–fatty acid cycle” or “Randle cycle” seems to occur. This involves a competition between glucose and fatty acids, where glucose oxidation is able to inhibit the conversion of triacylglycerol to long chain fatty acids (LCFAs) and LCFAs are able to inhibit glucose oxidation. The absence of significant alterations of TG levels corroborates this hypothesis. Among others, PFK has an important role in the extent of glycolysis inhibition [16]. This mechanism can be modulated by mTORC1, which is activated by insulin, growth factors, and amino acids to promote protein and sphingolipid synthesis, thus securing cell proliferation [17]. Previous studies reported a potential role of the estrogen receptors in this competition between glycolysis and fatty acid oxidation [18].

Also consistent across the molecular subtypes was the decrease in glutamine consumption. The association between glutaminolysis and tumor proliferative needs is well established, glutamine being the second most used energy substrate of tumor cells [19]. Glutamine is a substrate for nitrogen and carbon atoms. Tumor cell nitrogen is essential for the synthesis of non-essential amino acids and nucleotides. In turn, carbons are the major source of catabolic, anabolic, and amphibolic pathways in the cell, such as TCA cycle metabolites and in fatty acid synthesis [20]. The lower levels of released glutamine observed after exposure to adiposity environment in the four MS representative cell lines indicate that glutamine is being redirected for these synthetic pathways.

Another common feature is the significant increase in the cholesterol levels. Redundantly, cholesterol was already described as a risk factor in BC [21]. Internal cholesterol levels derived from the consumption of acetyl-CoA, was already described to activate the mTORC1 pathway [22], intrinsically associated with the “glucose–fatty acid cycle”. Moreover, cholesterol is also required for cellular membrane biosynthesis. Therefore, the significant cholesterol release observed after incubation with adipocyte secretome is probably caused by cholesterol synthesis from acetyl-CoA. Inversely, it is also likely that the large availability of this steroid in the adipocyte secretome may be taken up by the cancer cells to a larger extent than needed, then being released to the extracellular medium.

### 4.1. Effect of Obesity in Luminal A Representative MCF-7 Cell Line

When exposed to adipocyte secretome, no significant alterations were found in glucose, pyruvate, lactate, acetate, isoleucine, valine, and triglyceride levels in luminal A representative cell line. Contrarily to the other cell lines, acetate levels in extracellular medium did not differ from control in MCF-7. Nevertheless, cholesterol biosynthesis is significantly enhanced, implying that part of the acetate is driven to this steroid synthesis. In addition, the slight increase in pyruvate release is probably caused by the extensive consumption of alanine, which is transaminated to this keto acid. Part of this metabolite can either be used for acetyl-CoA synthesis in the mitochondria, thus keeping TCA cycle active, or result in fatty acid, membrane phospholipids, and cholesterol synthesis. Glutamine is significantly less consumed when subjected to adipocyte associated factors. This amino acid is rapidly converted into glutamate by glutaminase. The significant decrease of this latter amino acid in the medium suggests it is metabolized to α-ketoglutarate, entering the TCA cycle through an anaplerotic reaction as PFK was the only upregulated glycolytic enzyme. PFK catalyzes the phosphorylation of Fructose-6-phosphate to Fructose-1,6-bisphosphate in glycolysis, but it is also associated with several moonlight functions such as survival, apoptosis inhibition by BAD inactivation, alteration of cell shape and motility by binding to cytoskeleton, and transcription regulation [23,24]. We suggest that the upregulation of PFK can be attributed to the detected downregulation of TIGAR expression, since the decrease in TIGAR releases PFK control. Our results reveal that when a low increase in PFK is observed, TIGAR is prone to be upregulated whereas a high increase in PFK is associated with a downregulation of TIGAR. Given the overall results, we concluded that adiposity promotes a more anaplerotic profile in luminal A representative cell line MCF-7, attenuating glycolysis.

### 4.2. Effect of Obesity in Luminal B Representative BT-474 Cell Line

In control conditions, we observed that BT-474 cell line exhibits both a lower secretion of lactate, pyruvate, glutamate, and alanine, and a lower uptake of glucose, glutamine, isoleucine, and valine from culture media, when compared to the other cell lines. BT-474 is characterized by its high proliferation rates, which in turn leads to higher energetic/biomass requirements. We hypothesized that, given the high proliferative state, the majority of the metabolites are used by the cell, and not released to the extracellular medium. Similar results were described previously, namely with a decrease in glucose uptake compared to other cell lines [15]. We found a significant biological difference in acetate when compared to MCF-7, because while MCF-7 releases high amounts of this metabolite to the culture media, BT-474 cells consume it. Moreover, we observed a statistical difference regarding acetate consumption from culture media between adiposity and control conditions. We hypothesize that the higher consumption of acetate under exposure to adipocyte secretome is likely to be driven by the increased cholesterol synthesis observed under this condition. We also postulate that acetate can be further used for phospholipids synthesis given the proliferative needs of this cellular line [25]. The lower consumption of glutamine and concomitant increase in glutamate release to cell medium led us to propose that BT-474, contrarily to MCF-7, has a more cataplerotic profile.

Furthermore, almost all glycolytic and gluconeogenic enzymes were upregulated, although without statistical significance. This is the case with PCK and G6P. Both PCK and G6P function are dependent on the origin tissue (gluconeogenic or non-gluconeogenic tissue). In non-gluconeogenic tissues, as the case of BC, G6P is involved in cell cycle regulation, survival, and migration. The major described function of PCK is cataplerosis by enhancing TCA cycle intermediates’ disposal that are then used in other pathways, namely pyruvate synthesis, glyceroneogenesis, pentose phosphate pathway, and most importantly glutaminolysis [26]. Glutamate availability contributes to cancer metastasis through the upregulation of Rab27-dependent recycling of MT1-MMP to promote invasive behavior leading to basement membrane disruption [27]. Taken together, our data suggest that BT-474 luminal B representative cell line is more dependent of the carbohydrate metabolic pathways such as glycolysis and gluconeogenesis.

### 4.3. Effect of Obesity in HER2+ Representative SK-BR-3 Cell Line

Interestingly, we found that the metabolic influence of mature adipocytes in SK-BR-3 is similar to the one found in BT-474. One could speculate that HER2 could be the connection. Up until now, no specific ligand has been described for HER2, although it is known that HER2 is the most favorable dimerization partner for the three other members of the EGFR/ErbB family—1, 3 and 4 [28]. Several adipose tissue related factors such as Transforming Growth Factor α (TGFα) [3] may be involved in downstream proliferative pathways requiring the metabolic pattern found in our results. We found that acetate is consumed (in a statistically dependent manner), probably replenishing cholesterol synthesis pathway. Alanine release is significantly decreased, and glutamate is significantly increased. Since there is no difference in glutamine levels, glutamate comes from α-ketoglutarate in a cataplerotic reaction. Valine and isoleucine do not seem to play a significant role in these cells under the adipocyte secretome condition, presenting a moderate consumption.

SK-BR-3 was the only cell line overexpressing all glycolytic and gluconeogenic enzymes when exposed to adipocyte secretome, which is in agreement with the high proliferative profile of this cell line. PFK expression is somewhat increased, attributed to the TIGAR upregulation, as previously discussed.

### 4.4. Effect of Obesity in Triple Negative Representative MDA-MB-231 Cell Line

Triple negative representative cell line, MDA-MB-231, exhibited significant alterations in most metabolites studied. Glucose uptake and lactate production were decreased when exposed to adipocyte secretome, revealing a diminished Warburg effect. MDA-MB-231 also revealed the most accentuated acetate consumption correlated with a heightened cholesterol production. Alanine switched from release in control to uptake in adiposity mimicking conditions. The conversion of this amino acid into pyruvate is likely to occur, which then suffers oxidative decarboxylation originating acetylCoA. Upon adipocyte secretome exposure, this cell line does not depend on glutaminolysis, as we observed by the lower consumption of glutamine. Glutamate, in turn, originates a-ketoglutarate entering the TCA cycle by an anaplerotic reaction. Valine and isoleucine are significantly decreased in the MDA-MB-231 cell line. Elevated levels of these branched-chain amino acids (BCAA) are associated with body mass index, although no consensus has yet been reached regarding the potential risk in breast cancer [29]. Metabolically, both BCAA were already associated with glucose and lipid metabolism [30].

Concordant with the results obtained from BT-474 and SK-BR-3, MDA-MB-231 has an enzymatic pattern identical to MCF-7. Only PFK is increased, although without statistical significance, corroborating the unchanged TIGAR expression. HK, FBP, and G6P are significantly downregulated in accordance with the proposed attenuated dependence on glycolytic and gluconeogenic pathways.

Lastly, we would like to mention the study limitations. We are aware that cell culture medium from control and treatment are two different references with different nutrient composition. We overcome this by presenting the results as the difference between pre and post treatment. Additionally, we would like to highlight that: (1) we could not change the required medium for adipocyte differentiation since alterations would affect the protocol, and (2) we did not have access to adipose tissue from normal weight individuals. Moreover, we would like to mention that the use of only one representative line is not sufficient to extrapolate the results.

## 5. Conclusions

Our findings suggest that adipocyte-related factors impact breast tumor metabolism beyond estrogen/progesterone receptors. HER2 receptor seems to play a major role in central metabolism since molecular HER2 receptor positivity subtypes share common features that differ from those with HER2 negative receptors. Metabolic imbalance is more evident in triple negative BC representative cell line MDA-MB-231.

Overall, our results highlight the metabolic differences among molecular BC subtypes, relative to one representative cell line, but also the influence of the secretome of differentiated adipocytes on the central biochemical metabolism in each BC molecular subtype representative cell line. Further studies are needed regarding the importance of adipocyte-related factors in breast cancer development, which ultimately will provide new therapeutic guidelines.

## Figures and Tables

**Figure 1 cells-12-02123-f001:**
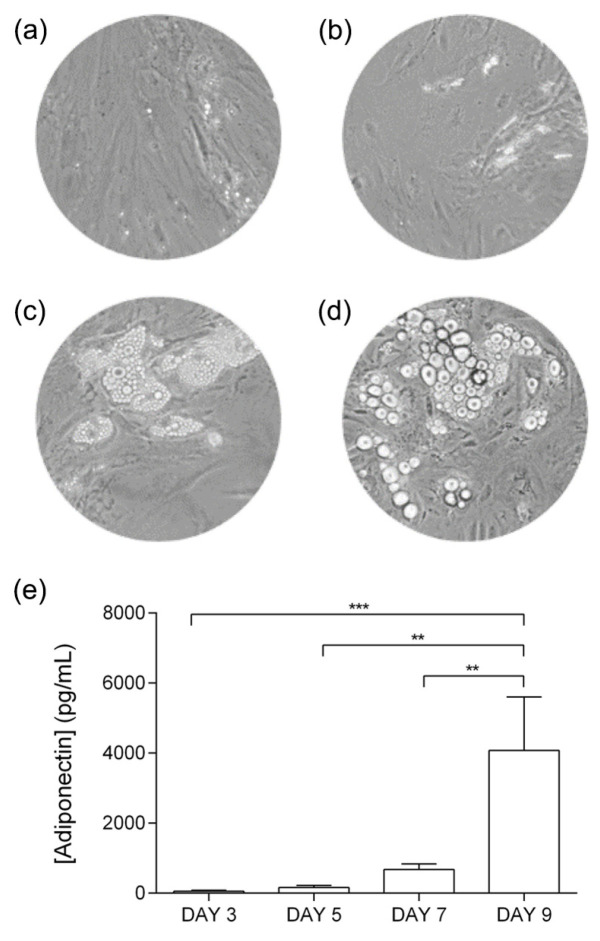
Phenotypic alterations of adipocyte differentiation at (**a**) day 3, (**b**) day 5, (**c**) day 7, and (**d**) day 9. Images were captured with 100x amplification. (**e**) Adiponectin quantification with a significant increase at day 9 (** *p*-value < 0.01; *** *p* < 0.001).

**Figure 2 cells-12-02123-f002:**
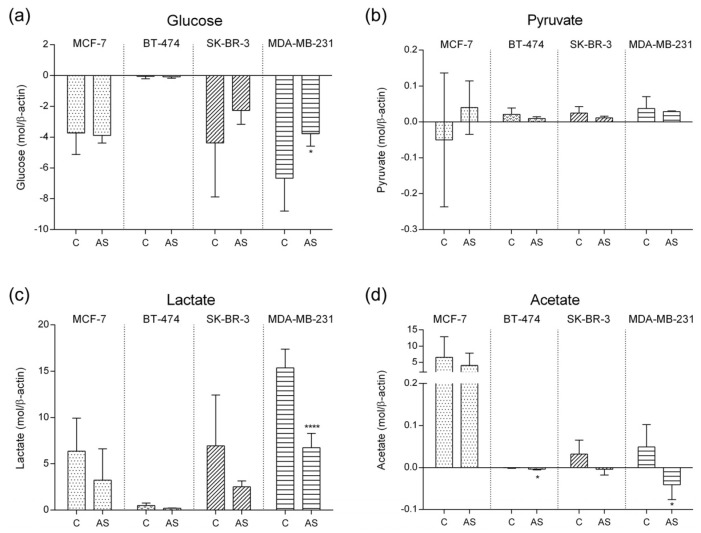
Metabolites concentration levels quantified in culture media by ^1^H–NMR normalized to β-actin expression stratified by molecular subtype representative cell line. (**a**) Glucose (*p*-value MDA-MD-231 = 0.0213); (**b**) Pyruvate; (**c**) Lactate (*p*-value MDA-MD-231 < 0.0001); (**d**) Acetate (*p*-value BT-474 = 0.0238; *p*-value MDA-MD-231 = 0.0133). (* *p*-value < 0.05; **** *p*-value < 0.0001). Positive values represent metabolite production and negative values represent metabolite consumption from the media. Legend: C—Control; AS—Adipocyte secretome.

**Figure 3 cells-12-02123-f003:**
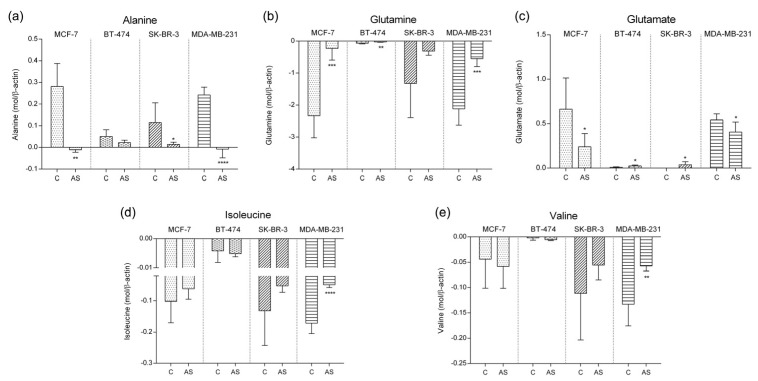
Amino acid concentration levels in culture media by ^1^H–NMR normalized to β-actin expression stratified by molecular subtype representative cell line. (**a**) Alanine (*p*-value MCF-7 = 0.001; *p*-value SK-BR-3 = 0.0366; *p*-value MDA-MD-231 < 0.0001); (**b**) Glutamine (*p*-value MCF-7 = 0.0003; *p*-value SK-BR-3 = 0.0097; *p*-value MDA-MD-231 = 0.0003); (**c**) Glutamate (*p*-value MCF-7 = 0.0377; *p*-value BT-474 = 0.0167; *p*-value SK-BR-3 = 0.0464; *p*-value MDA-MD-231 = 0.0495); (**d**) Isoleucine (*p*-value MDA-MD-231 < 0.0001); (**e**) Valine (*p*-value MDA-MD-231 = 0.0052). (* *p*-value < 0.05; ** *p*-value < 0.01; *** *p*-value < 0.001; **** *p*-value < 0.00001). Positive values represent metabolite production and negative values represent metabolite consumption from the media. Legend: C—control; AS—adipocyte secretome.

**Figure 4 cells-12-02123-f004:**
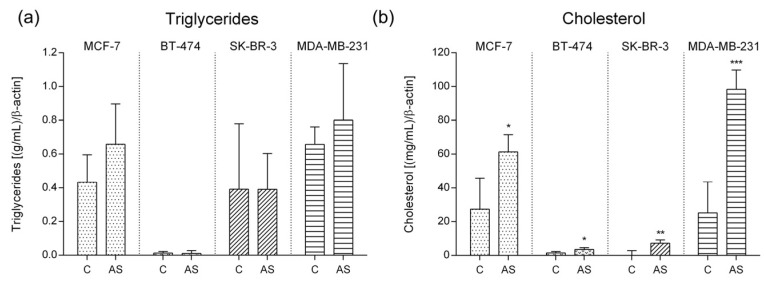
Lipid levels in culture media according to molecular subtype representative cell lines. (**a**) Triglycerides; (**b**) Cholesterol (*p*-value MCF-7 = 0.0179; *p*-value BT-474 = 0.0301; *p*-value SK-BR-3 = 0.0086; *p*-value MDA-MD-231 = 0.0005); (* *p*-value < 0.05; ** *p*-value < 0.01; *** *p*-value < 0.001). Legend: C—control; AS—adipocyte secretome.

**Figure 5 cells-12-02123-f005:**
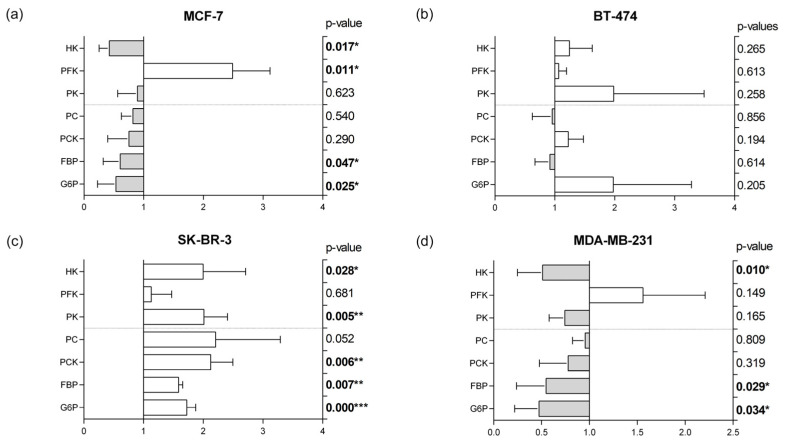
Enzymatic expression by RT-PCR of glycolytic and gluconeogenic patterns under adipocyte secretome exposure of the representative cell lines (**a**) MCF-7 for luminal A, (**b**) BT-474 for luminal B, (**c**) SK-BR-3 for HER2+, and (**d**) MDA-MB-231 for triple negative. Bars represent fold-increase relative to control; Positive fold-increase in white bars and negative fold-increase in grey bars. Legend: * Statistical significance highlighted in bold; Significance levels: * *p* < 0.05; ** *p* < 0.01; and *** *p* < 0.001; HK—Hexokinase; PFK—Phosphofructokinase; PK—Pyruvate kinase; PC—Pyruvate carboxylase; PCK—Phosphoenolpyruvate carboxykinase; FBP—Fructose-1,6-bisphosphatase; G6P—Glucose-6-phosphatase.

**Figure 6 cells-12-02123-f006:**
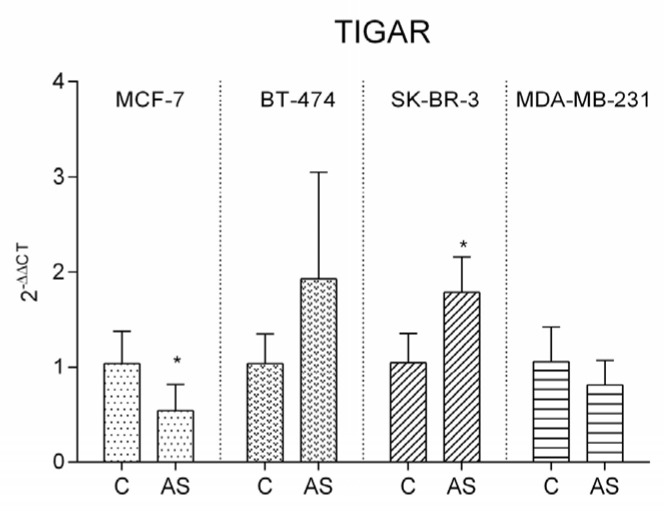
TP53 Induced Glycolysis Regulatory Phosphatase (TIGAR) enzymatic expression in each representative cell lines by RT–PCR (*p*-value = 0.0475 for MCF-7; *p*-value = 0.0161 for SK-BR-3); (* *p*-value < 0.05). Legend: C—control; AS—adipocyte secretome.

**Table 1 cells-12-02123-t001:** Specifications of cell lines used, representative of each molecular subtype [11].

Cell Line	Type	Molecular Subtype	Receptor Status
MCF-7	Invasive Ductal Carcinoma	Luminal A	ER+ PR+ HER2−
BT-474	Invasive Ductal Carcinoma	Luminal B	ER+ PR+ HER2+
SK-BR-3	Adenocarcinoma	HER2+	ER− PR− HER2+
MDA-MB-231	Adenocarcinoma	Triple Negative	ER− PR− HER2−

Legend: ER—estrogen receptor; PR—progesterone receptor; HER2—human epidermal growth factor receptor 2.

**Table 2 cells-12-02123-t002:** Primer sequences used in the RT-PCR analysis.

Gene	Primer Sequence (5′-3′)
ACTB *	F—GACGACATGGAGAAAATCTG R—ATGATCTGGGTCATCTTCTC
HK-2	F—CTAAACTAGACGAGAGTTTCC R—CATCATAACCACAGGTCATC
PFK-L	F—GAAAGACACTGATTTCGAGC R—CATACTGATGCGGTATTGTG
PK-L/R	F—AAAATTGAGAACCACGAAGG R—CAATCATCATCTTCTGAGCC
PC	F—TTATGGTGCAGAATGGATTG R—TCAGTACCTTAGAGCGAAAG
PCK-2	F—GGCATTCGAGATTTTGTAGAG R—CAGCAGTTATTGTACTTGGG
FBP-1	F—CCAGATAATTCAGCTCCTTATG R—AGAAATATCCCTCCGTAGAC
G6P	F—ACTGTGCATACATGTTCATC R—TGAATGTTTTGACCTAGTGC

Legend: F—Forward; R—Reverse; *—Housekeeping gene; ACTB—β-actin; HK2—Hexokinase II; PFK-L—Phosphofructokinase—Liver; PK-L/R—Pyruvate Kinase—Liver and red blood cells; PC—Pyruvate Carboxylase; PCK-2—Phosphoenolpyruvate Carboxykinase—2; FBP-1—Fructose-1,6-bisphosphatase—1; G6P—Glucose-6-Phosphatase.

## Data Availability

The data presented in this study are available on request from the corresponding author.

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
