# Peer review of "The Influence of Adipocyte Secretome on Selected Metabolic Fingerprints of Breast Cancer Cell Lines Representing the Four Major Breast Cancer Subtypes"

_cells, 2023, doi:10.3390/cells12172123_

Round 1
Reviewer 1 Report
Comments:
Abstract:
1. Line 29: Please remove h from the beginning of this senrtence “hThe present work aims to elucidate the metabolic alterations in different MS representative cell lines subjected to obesity mimicking conditions.”
Introduction:
1. Line 57: “On the other hand, luminal B usually expresses decreased levels of 57 ER as well as PR, elevated levels of proliferation markers (such as Ki67) and of cell cycle-58 associated genes (like G2/mitotic-specific cyclin B1 – CCNB1, and MYB Proto-Oncogene 59 Like 2 – MYBL2) and may or not express HER2[2]. “
This sentence is unclear. Relative to what is the level of ER and PR decresed, while proliferation level elevated?
Materials and methods
1. Line 109: “Cells were incubated with DMEM/F12 medium supplemented with 17.5 mM of glucose, 1% penicillin/streptomycin, 18 μM of pantothenic acid, 110 100 μM of ascorbic acid, 16 μM of biotin, 3% NCS, 10 μg/mL of insulin, 0.1 μM of dexamethasone until the fully differentiated phenotype was reached.”
How it was estimated that the fully differentiation of adipocytes was reached? were the cells incubated with the differentiation medium for the same number of days each time or not?
2. Line 113: “Pool of adipocyte secretomes was collected from 7 assays.”
What does it mean? That Authors collected conditioned media from adipocytes derived from 7 different patients? After what time was the medium collected and treated as conditioned medium? Were media collected from adipocytes’ derived from different patients and pooled together?
3. Line 117: “Adiponectin was measured through a commercial ELISA kit to access differentiation grade.”
On what basis adiponectin was considered to be a sufficient marker to determine whether adipocytes are fully differentiated? Why only one marker was used?
4. Were the cell lines used bought from commercial sources or authenticated?
5. Line 128: “Serum-free DMEM was used as control and treatments were performed using the adipocyte differentiation secretome for 24 hours.”
Since as the control were treated cells incubated in DMEM without FBS, I understand that the conditioned media was also free of FBS. Did the cancer cells have enough nutrients in such a medium, since the nutrients present in the medium were, at least in part, consumed by the adipocytes that were cultured in this medium previously?
6. What does the phrase "adipocyte differentiation secretome" mean? maybe "secretome of differentiated adipocytes" would sound better
7. Why beta actin was chosen as an internal control for NMR and RT-PCR method? The level of this gene expression is often changed in cancer cells (e.g. “ACTB in cancer”
Guo C, Liu S, Wang J, Sun MZ, Greenaway FT. Clin Chim Acta. 2013 Feb 18;417:39-44. doi: 10.1016/j.cca.2012.12.012.)
Can the Authors show the expression of beta actin on mRNA level in all cell lines they tested and compare the results with each other?
8. Line 154: “Triglyceride quantification was determined by enzymatic colorimetric assay using standard methods, on the AU 5800 analyzer (Beckman Coulter®) according to the manufacturer instructions and internal patronized protocol.”
The methods should be described in such a way that the reader is able to reproduce them in his own lab. This description does not allow us to say how the experiment was conducted - it does not even include the name of the manufacturer of the used kit
9. In Line 130 Authors say "Assays were performed in quintuplicate", while the RT-PCR description on line 175 they say "Assays were performed in duplicate". Which version is correct? Please clearly state which experiments were performed in which number of repetitions.
Results:
1. In my opinion the nomenclature used in the work to describe the tested samples is incorrect. Since the control are just the breast cancer cells, I would not describe conditions of this co-culture as "normoponderal mimicking conditions". Adipocytes are present in the organism physiologically in the vicinity of breast cancer cells, also in the case of lean people. The treated cells were cultured in the presence of adipocyte conditioned medium, but we can’t name these conditions "adiposity mimicking ones". These cells were just treated with the conditioned medium from adipocytes. In my opinion, if pre-adipocytes were isolated from lean people and breast cancer cells were cultured in the presence of medium from these adipocytes, it could be called "normoponderal mimicking conditions", while if pre-adipocytes were isolated from obese people, and breast cancer cells were cultured in the presence of medium from these adipocytes, then it would be possible to call it "adiposity mimicking conditions”.
Were the used preadipocytes derived exclusively from obese people? Even so, control cells still need to be treated with adipocyte media from lean individuals to be able to use the above mentioned names of conditions. For the same reason, I also suggest changing the title of the paper - it does not concern the influence of obesity, but the influence of the adipocyte secretome on metabolic fingerprints of in vitro breast cancer subtypes.
2. Line 191 “We then used day-9 secretome in the next experiments”. The classic protocol (e.g. Optimization of the differentiation of human preadipocytes in vitro.
3. Hemmrich K, von Heimburg D, Cierpka K, Haydarlioglu S, Pallua N. Differentiation. 2005 Feb;73(1):28-35 ) assumes 14 days of differentiation - why were the preadipocytes differentiated for only 9 days?
4. Line 199: “The levels of glucose as well as glycolysis end products, pyruvate and their dual metabolites, lactate, and acetate can be visualized in Figure 2.”
The word visualized in this sentence is not used in the correct context.
5. Line 198: What kind of media was used for this experiment? Control media from cancer cells and media from cancer cells after treatment with adipocyte conditioned media? This information should be included in the manuscript.
6. Line 221” Altogether, these findings suggest that under an obesity environment, the cell lines representative of the four MS of BC present distinct metabolic pathways.”
Authors should avoid writing about obesity, because using the described conditions of the experiments they do not study obesity. Moreover, I would soften this statement - only one line from each subtype was tested - it is difficult to draw general conclusions on this basis.
7. Line 304: “Given the results obtained in PFK expression, we further quantified the expression of TP53 Induced Glycolysis Regulatory Phosphatase (TIGAR), an enzyme that modulates glycolysis by targeting PFK”
Why was it decided to investigate TIGAR? After all, PFK expression changes significantly only in the case of the MCF7 line.
Discussion:
1. For the reasons mentioned above, I believe that in the discussion, the Authors should not describe the effect of the adipocyte-conditioned medium on cells as the effect of obesity.
Conclusions:
1. Line 457 “Overall, our findings are of extreme importance, not only to highlight the differences 457 among molecular BC subtypes but also the influence of obesity on the central biochemical 458 metabolism in each BC molecular subtype”
This sentence is exaggerated as the Authors do not study the effect of obesity on cells, and conclusions should be drawn more carefully, because they are drawn on the basis of examining only one line representing each subtype.
In my opinion the quality of English Language is acceptable.
Reviewer 2 Report
An interesting paper that I enjoyed reading.
I had some questions mainly about the methodology as the interpretation of the results may be slightly different depending on the protocols etc.
· Cells were exposed to secretome for 24 hrs (from a pool of 7)-
When was the secretome collected- always on day 9 or when the same level of differentiation had been reached?
Was it added neat ? When removed what was it replaced with or was the experiment concluded after 24hrs? Was the control cell media in the consequent experiments on the cancer cells, the same as the differentiation media?
After the 24hr exposure was the cell phenotype assessed? -e.g.had the cells grown, look different? Important to comment on as will be different basally even between cell lines.
Was there any possibility that the nutrients in the secretome mixture (in whatever media) had been depleted by the cancer cells more in one cell line over another, particularly as they grow at different rates.
· Line 189- authors indicated day 9 had the highest differentiation rate-
This seems quite early on in the process- did the authors test for longer?
Were any other markers of differentiation used, such as Pref-1?
· I am not familiar with presentation of metabolite data- so could the authors indicate why the levels were corrected for beta-actin m RNA levels? Were the cholesterol measurements in the supernatants corrected for anything?
· I agree with most of the conclusions, but I think in places the conclusions should be tempered as many of the results were not significant-I appreciate that trends, however, were visible.
Minor
-Extra ‘h’ in line 29 before The
-Line 38, ‘disclosure’ choose different word- maybe ‘indicate’
In general editing is needed of the English as the use of some words is incorrect for the desired meaning.
Round 2
Reviewer 1 Report
I have to admit that the Authors made many corrections to the manuscript. However, one issue raises my serious doubts. If I understand correctly, the control cells were cultured in DMEM medium and the adipocytes in DMEM/F12. Then, the medium above the adipocytes was collected and the tested cells were cultured in it. So how can we compare control cells to treated cells if they were grown in two different types of medium? Especially considering the fact that in the NMR experiments the media in which the cells were grown were analyzed in terms of the metabolites contained in them. If DMEM and DMEM/F12 initially differ in the content of glucose, individual amino acids, pyruvate, etc., the amounts of metabolites in control cells and those treated with adipocyte media cannot be reliably compared in this assay system. I am asking the authors to compare the composition of both media and specify the differences between them.
Reviewer 2 Report
It appears that the basic composition of the media between the controls cells and those treated with secretome was different. The authors clearly cant repeat everything but I do feel this is a limitation that should be acknowledged in the discussion.
